# Comprehensive analysis of the circadian nuclear and cytoplasmic transcriptome in mouse liver

**Clémence Hurni**, **Benjamin D. Weger**, **Cédric Gobet**, **Felix Naef** *

The Institute of Bioengineering (IBI), School of Life Sciences, Ecole Polytechnique Fédérale de Lausanne (EPFL), Lausanne, Switzerland

* felix.naef@epfl.ch

## Abstract

In eukaryotes, RNA is synthesised in the nucleus, spliced, and exported to the cytoplasm where it is translated and finally degraded. Any of these steps could be subject to temporal regulation during the circadian cycle, resulting in daily fluctuations of RNA accumulation and affecting the distribution of transcripts in different subcellular compartments. Our study analysed the nuclear and cytoplasmic, poly(A) and total transcriptomes of mouse livers collected over the course of a day. These data provide a genome-wide temporal inventory of enrichment in subcellular RNA, and revealed specific signatures of splicing, nuclear export and cytoplasmic mRNA stability related to transcript and gene lengths. Combined with a mathematical model describing rhythmic RNA profiles, we could test the rhythmicity of export rates and cytoplasmic degradation rates of approximately 1400 genes. With nuclear export times usually much shorter than cytoplasmic half-lives, we found that nuclear export contributes to the modulation and generation of rhythmic profiles of 10% of the cycling nuclear mRNAs. This study contributes to a better understanding of the dynamic regulation of the transcriptome during the day-night cycle.

## Author summary

### Why was this study done?

- The circadian clock coordinates temporal gene expression programs to synchronise cellular and cellular functions with environmental time.

- The circadian clock impinges on many layers of gene regulation, from transcription to mRNA degradation in the cytoplasm.

- Specifically, whether temporal modulation of nuclear export could influence rhythmic mRNA expression was not known.

**Data Availability Statement:** All RNA-sequencing files are available from the Gene Expression Omnibus database (accession number: GSE182856)

**Funding:** This work was supported by the Swiss National Science Foundation (SNSF) project grants 310030_173079 and 310030B_201267 (to F.N.), and the EPFL. The funders had no role in study design, data collection and analysis, decision to publish, or preparation of the manuscript.

**Competing interests:** The authors have declared that no competing interests exist.

## What did the researchers do and find?

- We sequenced RNA from nuclear and cytoplasmic fractions from mouse livers sampled at different times of the day.

- Using a mathematical model describing temporal accumulation profiles of pre-mRNA, nuclear mRNA and cytoplasmic mRNA, we could estimate the nuclear export rates and cytoplasmic degradation rates of approximately 1400 genes.

- We observed rhythmic patterns in the nucleus that most likely originate from a rhythmic regulation of the nuclear export rate, affecting approximately 10% of the oscillations of the nuclear transcriptome in the liver.

## What do these findings mean?

- Our fractionated RNA analysis contributes to a better understanding of the dynamic regulation of the transcriptome during the daily cycle.

- The majority of the RNA transcripts are exported to the cytoplasm in less than 30 minutes, however, some are retained in the nucleus for several hours. Thus, cellular fractionation allows a finer resolution of temporal profiles in the different subcellular compartments, which is otherwise masked at bulk-level.

## Introduction

Every step of the RNA life cycle is tightly regulated to ensure proper cellular function, including transcription, co- and post-transcriptional splicing, nuclear export, and cytoplasmic degradation. Collectively, the balance of production and elimination rates determines the abundance of RNA species (e.g. pre-mRNA, polyadenylated pre-mRNA, mRNA) in a given subcellular compartment (e.g. nucleus, cytoplasm) [1]. Studies based on snapshot images of subcellular RNA distributions, either obtained by single-molecule RNA-FISH (smFISH) [2], or by RNA-seq of fractionated cells, allowed to characterise nuclear and cytoplasmic transcriptomes [3–6], revealing diverse patterns of subcellular localisation for different RNA subclasses. Interestingly, a study in the mouse liver showed that approximately 10% of protein coding transcripts were retained in the nucleus, efficiently buffering cytoplasmic noise associated with stochastic transcription [4].

RNA processing rates, and particularly mRNA degradation rates have been estimated in cells either from transcriptional blockage experiments [7] or a pulse-chase strategy to label nascent transcripts during a short period of time [8]. To estimate life times, RNA levels are typically fitted to exponential functions. Alternatively, mathematical modelling using ordinary differential equations to jointly describe pre-mRNA and mRNA levels can indirectly infer kinetic parameters of RNA processing steps in a label-free and genome-wide manner [9]; however, dynamic datasets reflecting different cellular states are needed to identify the parameters, for instance during cell cycle, cell differentiation [10], or at different circadian time-points [11,12]. Our laboratory previously developed a mathematical model describing the temporal profiles of pre-mRNA and mRNA from total RNA-seq, which uses the periodicity of circadian

cycles and the relationship between peak times and relative amplitudes to infer the production and the decay terms; this study allowed the identification of hundreds of post-transcriptionally generated rhythmic mRNA profiles [12].

Among mouse tissues, the liver has the highest number of cycling mRNA [13], and the rhythmic patterns are driven by the endogenous circadian clock as well as by the fasting/feeding cycle [14–16]. The circadian clock is a self-sustained transcription-translation feedback oscillator [17,18]. In mammals, the network consists of the CLOCK/BMAL1 transcription factors dimer which promotes the expression of its own repressors Cryptochromes (CRY1 and CRY2) and Periods (PER1 and PER2), generating cycling negative feedback loops with a period of approximately 24 hours. A second feedback loop is composed of the complex CLOCK/BMAL1 activating the expression of target genes such as *Rev-Erbα,β*, which repress the expression of BMAL1. The molecular clock also targets thousands of downstream *clock-controlled genes* to couple circadian cycles and physiological functions [19,20].

In the circadian context, rhythmic transcription often underlies rhythmic gene expression, and has been extensively studied in mouse liver [21–23]. Interestingly, rhythmic transcription does not account for the rhythmic accumulation of all mRNAs, implying regulation at the post-transcriptional level. Indeed, the rates regulating the RNA processing steps often adapt to external inputs such as temperature shifts [24], nutrient levels [4], inflammation [5], or time of the day [11,12]. In principle, every step during the RNA life-cycle could be regulated temporally, contributing to rhythmic gene expression. The extent of which post-transcriptional regulation contributes to circadian gene expression is debated, varying from 15% [21,25] to 70% [22]. Conversely, many rhythmically transcribed genes do not have corresponding cycling mRNA [22] or protein [26], suggesting that the oscillatory amplitudes are damped as rhythm propagates from early to later steps of gene expression [27].

This study aims at understanding the circadian dynamics of RNA regulatory steps in mouse liver. More specifically, we focus on nuclear export and cytoplasmic degradation. We performed RNA-sequencing of polyadenylated and ribo-depleted total RNA purified from nuclear and cytoplasmic fractions of mouse liver samples collected every 4 hours along a full day cycle. This dataset provides a genome-wide and temporal inventory of RNA levels in two subcellular compartments. At the static (non-temporal) level, we found a relationship between the transcript length and the localisation, with long transcripts being enriched in the nucleus. Additionally, gene length influences the proportion of co- and post-transcriptional splicing. By modelling the rhythmic profiles (amplitudes and phases) of nuclear pre-mRNA, nuclear mRNA, and cytoplasmic mRNA, we estimated the export rates and cytoplasmic degradation rates of approximately 1400 genes. We additionally uncovered rhythmic patterns in the nucleus that suggest a rhythmic nuclear export. Moreover, this post-transcriptional regulation step not only contributes to amplifying the amplitude of highly rhythmic genes but also generates *de novo* rhythms, and is predicted to influence the temporal profiles of approximately 10% of the rhythmic nuclear transcriptome.

## Results

### Global characteristics of Poly(A) and Total RNA populations in hepatic nuclear and cytoplasmic fractions

To characterise the circadian kinetics of RNA regulatory processes occurring in different cellular compartments, we isolated RNA from nuclear (Nuc / N) and cytoplasmic (Cyt / C) fractions from mouse livers sampled every 4 hours during two consecutive days (Fig 1A). We sequenced ribosome-depleted total RNA (T) and polyadenylated RNA (A). As a control, we also extracted RNA from unfractionated samples (Unf / U). We quantified the transcript levels

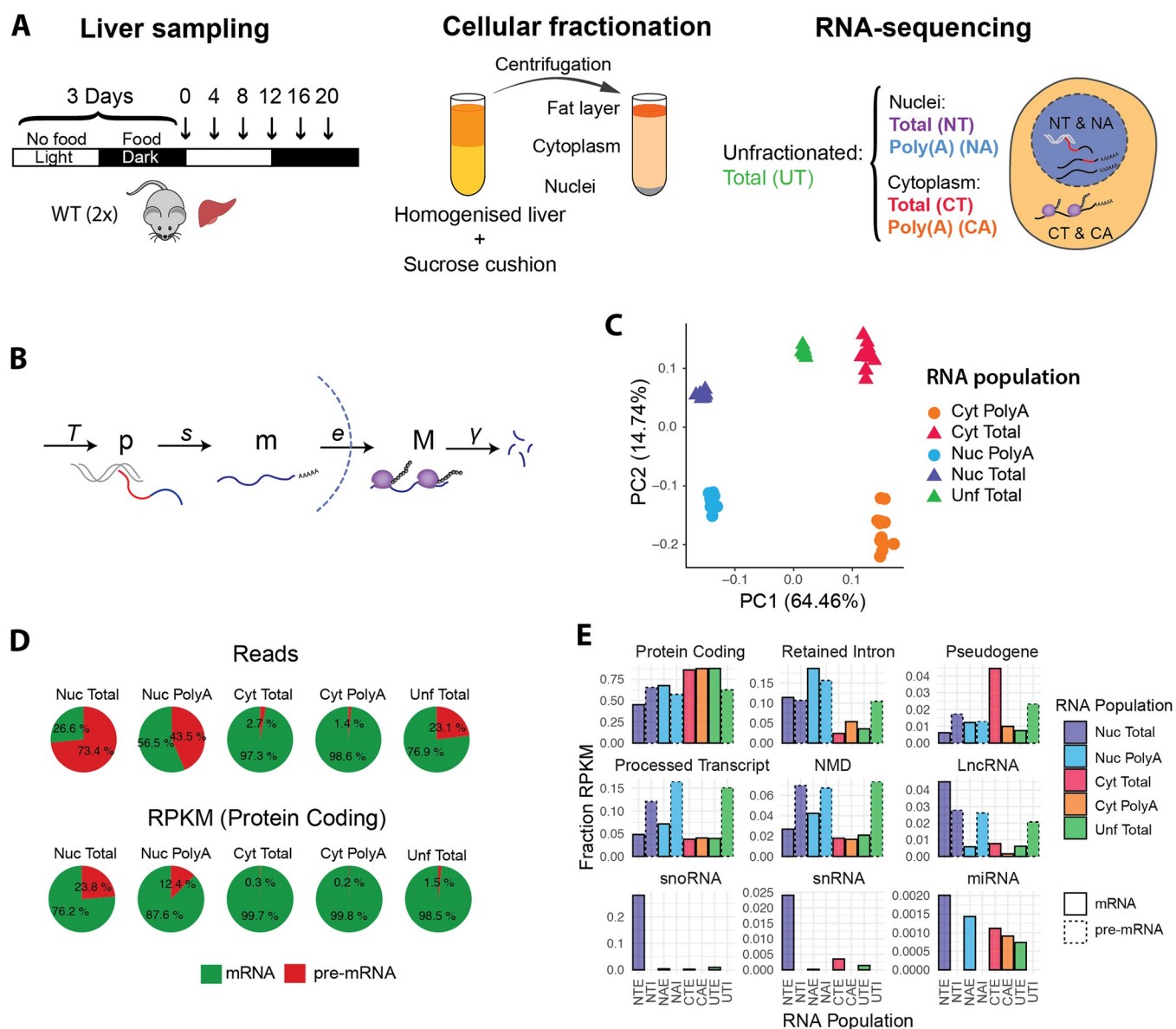

**Fig 1. Global characteristics of Poly(A) and Total RNA populations in nuclear and cytoplasmic fractions. A**: Experimental design: 12 WT mice were kept in a 12:12 light-dark cycle. Food was given only during the dark phase (active phase). The night-restricted feeding regimen started 3 days prior to the sampling. Livers were collected every 4 hours (n = 2), homogenised, and centrifuged in a sucrose cushion as described in [23]. RNA from cytoplasmic and nuclear fraction were extracted. rRNA-depleted Total RNA (T) and Polyadenylated RNA (A) from nuclear and cytoplasmic fractions were sequenced. Total RNA from unfractionated liver tissues was also sequenced as an internal reference set. **B**: Representation of each RNA processing step modelled in this study: starting with the transcription by RNA Polymerase II (at a rate *T*), splicing of the introns (red segment) and addition of a poly(A) tail at the 3' end (*s*), export to the cytoplasm (*e*), and degradation (*γ*). *p* = pre-mRNA, *m* = nuclear polyadenylated mRNA, *M* = cytoplasmic mRNA. **C**: Principal Component Analysis using mRNA (signal from exonic regions only). PC1 separates nuclear from cytoplasmic samples, and PC2 separates Total from Poly(A) samples. Each datapoint represents one time point. **D**: Percentage of reads mapping on pre-mRNA or mRNA transcripts in each condition (up). Proportion of RPKM (reads normalised by gene length) of protein coding transcripts mapping on pre-mRNA and mRNA (bottom). **E**: Fraction of each biotype in each RNA sample. Solid lines represent mRNA (first bar), dotted lines represent pre-mRNA (second bar). Pre-mRNA from cytoplasmic fractions are not shown.

using the pseudo-alignment algorithm implemented in Kallisto [28] (S1 Table), by providing as a reference the annotation of transcript sequences (isoform) of both pre-mRNA (intronic and exonic regions, labelled I) and mRNA (exonic regions only, labelled E). Nuclear Total pre-mRNA (NTI), Nuclear Poly(A) mRNA (NAE), and Cytoplasmic Poly(A) mRNA (CAE) were

used in our simplified model describing the sequential events occurring along the RNA life-cycle (Fig 1B). This includes the steps of pre-mRNA processing (splicing + polyadenylation), export of nuclear mRNA, and degradation of cytoplasmic mRNA. We reduced the complexity of the RNA-seq quantification performed at the transcript-level by either summing the counts of all the expressed isoforms of a gene, or of all the isoforms belonging to the same biotype (S1 Table, see Methods). The counts estimated by Kallisto were further normalised by the gene length to obtain a quantification akin to reads per kilobase per million (RPKM) (S2 and S3 Tables, see Methods). Finally, we calibrated the nuclear-to-cytoplasmic ratio using a previously published dataset, where RNA-seq counts were scaled based on smFISH quantifications of mRNA copies per cell [4].

Principal Component Analysis (PCA) on the mRNA measurements shows that the first two components readily split the types of samples: indeed, subcellular localisation explains most of the variation between samples (64%, Fig 1C). Unfractionated samples lie between nuclear and cytoplasmic samples on the first principal component (PC1), although closer to cytoplasmic RNA. PC2 accounts for 15% of the variability, which is mainly related to differences in the type of RNA (Poly(A) versus Total). In comparison, the temporal variation is significantly smaller (PC4 and PC5, < 1.5% each).

The Nuclear Total RNA population (NT) includes all the transcripts found in the nucleus, from nascent transcripts to fully transcribed polyadenylated mature RNAs. As expected, the proportion of intron versus exon reads was the highest in the NT sample, and lowest in the cytoplasmic fraction (Fig 1D). In terms of reads (reflecting the mass of the corresponding RNA), approximately 75% of the reads align to pre-mRNA sequences. These proportions are similar to what has been previously described in mouse liver for nuclear total RNA [29]. Nuclear Poly(A) RNA population (NA) represents the population of fully-transcribed and polyadenylated transcripts. Interestingly, 45% of these reads map on pre-mRNA (Fig 1D). When reads are normalised by the gene length (RPKM), this showed that about 1 in 8 polyade-nylated transcripts in the nucleus were still not fully spliced. Note that in Fig 1D, we only reported fractions of protein coding transcripts to avoid biases associated with the different compositions of RNA biotypes in different fractions (Fig 1E). Indeed, even if most of the detected RNA are protein coding transcripts, making up to 85% of mRNA molecules in the cytoplasm and in the whole cell, nuclear RNA populations are more diverse. The nucleus is particularly enriched for functional non-coding RNAs such as snoRNA, and for isoforms with a retained intron (10 to 20% of the nuclear RNA fraction). Finally, as expected, pre-mRNA represents less than 3% of the cytoplasmic RNA fraction.

In light of our results, we conclude that our method of fractionating liver cells performed as expected, and that our analysis of RNA distribution in the nucleus and cytoplasm supports previous findings [3–6,30].

## Static relationships between levels of successive RNA species and dependencies on cytoplasmic RNA stability, nuclear export and splicing rates

Before analysing temporal aspects of these datasets, we first studied the relationships between the temporal averages of the various RNA species, in particular properties related to cyto-plasmic RNA stability, nuclear export, and splicing.

First, we compared nuclear (NAE) and cytoplasmic (CAE) polyadenylated mRNAs. As noted in a previous study [4], in the simplest scenario where there is no degradation in the nucleus, the steady-state ratio of NAE over CAE reflects the ratio of two parameters: the nuclear export rate and the cytoplasmic degradation rate (Fig 1B). A fast export from the

nucleus to the cytoplasm would deplete the amount of RNA in the nucleus, and a long cytoplasmic half-life in the cytoplasm would lead to an accumulation of mRNA in the cytoplasm. On the contrary, RNAs with a long export time (or long nuclear retention) and short half-life in the cytoplasm would increase the nuclear to cytoplasmic ratio, resulting in an apparent nuclear accumulation.

Across all genes, the expression levels of NAE span 8.8 $\log_2$ RPKM ($10^{2.6}$ fold). CAE has a wider distribution of expression level of 12.5 $\log_2$ RPKM (about $10^4$ fold, Fig 2A). The most abundant transcripts code for secreted proteins such as Murine Urinary Proteins, Albumin, and Apolipoproteins, and are enriched in the cytoplasm. The NAE/CAE ratio varies by a factor of almost 2500, with a median $\log_2$(NAE/CAE) around -2.3, indicating that RNA transcript counts are more abundant in the cytoplasm than in the nucleus. The nuclear-enriched RNA population includes the long non-coding RNAs *Malat1* and *Neat1*, which form the backbones of nuclear speckles and nuclear paraspeckles [31,32]. Moreover, a fraction of transcripts are also found enriched in the nucleus, consistently with a previous study [4] showing that in the mouse liver, protein coding genes such as *Nlrp6* (here $\log_2$-ratio = 1.6), and glucose metabolism-related genes *Mlxipl* ($\log_2$(NAE/CAE) = 3.4), and *Gcgr* ($\log_2$(NAE/CAE) = -0.1) were indeed enriched in the nucleus.

In our simplified model, the steady-state $\log_2$(NAE) = $\log_2$(Transcription rate) + $\log_2$(export time), and $\log_2$(CAE) = $\log_2$(Transcription rate) + $\log_2$(half-life). Note here that our system is underdetermined, and that times / rates discussed here are relative. The fact that $\log_2$(NAE) and $\log_2$(CAE) strongly co-vary, and that the transcription rate is added to both, implies that this process carries the largest variation, as expected. We next aimed to determine which molecular process co-varied most with the NAE/CAE ratio. The strong correlation coefficient with NAE and absence of correlation with CAE suggests that the cytoplasmic half-life has a higher influence on the ratio than the export time (Fig 2B). We verified the subcellular localisation of a known nuclear-biased transcript (*Mlxipl*) by smFISH, and its nuclear accumulation was evident, although not precisely quantifiable due to its high abundance and crowdedness of mRNA molecules (Fig 2C). On the opposite side, *Actb* is a typical example of a cytoplasmically enriched transcript ($\log_2$(NAE/CAE) = -3.64), which was verified by smFISH, showing a $\log_2$(NAE/CAE) that varied between time points, ranging from -1.42 to -3.8, with a median of -2.7.

To identify the biological functions of nuclear and cytoplasmic-enriched RNA populations, we performed an ontology gene set functional analysis on nuclear and cytoplasmic-biased protein coding genes. We found that transcripts coding for histone modifiers and mRNA processing (splicing and export) are enriched in the nucleus, while proteins encoded by cytoplasmic-enriched RNA were involved in translation and ribosome biogenesis, cellular respiration, sterol metabolism (Fig 2D and S4 Table).

We next examined several genomic features that may modulate kinetics (transcription rate, export, decay) and as a consequence mRNA localisation, such as gene length, transcript length, number of exons, 3'UTR and 5'UTR length [6,30,33]. The 3'UTR length (but not the 5'UTR length) showed a positive correlation with the nuclear to cytoplasmic ratio ($R^2 = 0.16$), indicating that transcripts with longer 3'UTR sequences are enriched in the nucleus (Fig 2E). The length of the 3'UTR is often proposed as a predictor of mRNA stability, because of the presence of many regulatory elements. However, both stabilising and destabilising RNA-Binding Proteins and miRNAs often compete for the same mRNA substrate [34], which in the end results in a conflicting or undetectable effect of the 3'UTR length on processing rates [6]. Strikingly, we found that spliced transcript length (corresponding to the total exon length) has a significant positive correlation with nuclear localisation ($R^2 = 0.31$, Fig 2F), and the correlation is improved when focusing only on protein coding transcripts (Fig 2G, $R^2 = 0.41$). As transcripts of hundreds of kilobases take several hours to be fully transcribed [35], this could explain the

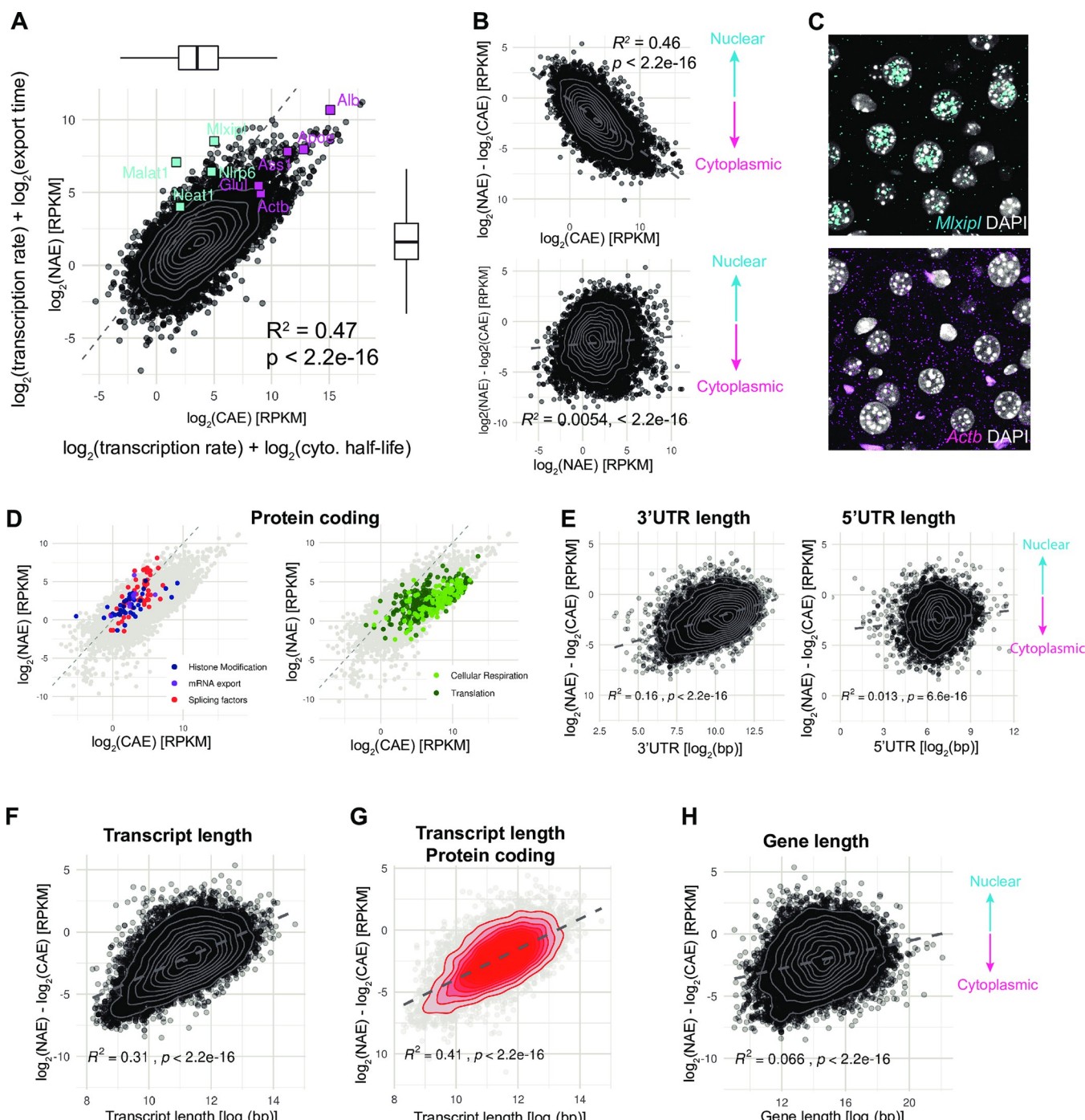

**Fig 2. The ratio of nuclear and cytoplasmic mRNA abundance is related to transcript length. A:** Expression level of nuclear mRNA (NAE) versus cytoplasmic mRNA (CAE) in $\log_2$(RPKM), averaged over 6 time-points and 12 animals. In our model, the level of NAE corresponds to the transcription rate plus the nuclear half-life, and the level of CAE to the transcription rate plus cytoplasmic half-life (all in $\log_2$-scale). Cyan squares represent known nuclear genes and magenta squares cytoplasmic genes [4]. Dashed line indicates the 1:1 ratio. Note here that the diagonal line has been scaled with the dataset (see Methods). Grey contour lines: 2D kernel density estimation. Number of genes = 10914. **B**: $\log_2$-ratio of NAE / CAE against the average expression in the cytoplasm in $\log_2$-scale (up) or in the nucleus (bottom). **C:** smFISH of the *Mlxipl* and *Actb* transcripts in liver FFPE sections, at ZT12. Nuclei are stained with DAPI (grey). **D**: GO Terms enrichment analysis of nuclear and cytoplasmic protein coding genes. Left: All the genes belonging to the nuclear enriched GO Terms are highlighted in color: histone modifications (GO:0016570, GO:0033169, GO:0070076, GO:0006306, and GO:0044728), mRNA export (GO:0010793, GO:0010793), and splicing (GO:0006376, GO:0043484). Right: all the genes belonging to some of the cytoplasmic enriched GO Terms are highlighted in color: Cellular respiration (GO:0042775, GO:0022904, GO:0015986, GO:0006123), and Translation (GO:0006412, GO:0002181, GO:0006414,). **E, F, G, H**: $\log_2$(NAE/CAE) ratio against the following genomic features: 5'UTR length and 3'UTR length (E), transcript length (spliced isoform, F, G) and gene length (exon

+ intron, H). To simplify, these transcript-specific features were averaged (weighted) based on the expression levels of each transcript to obtain a value per gene (E, F, H) or per biotype (G). Grey dashed line is the regression line.

longer residence time in the nucleus. However, in that case, we would expect a correlation with gene length as well, which was not observed (Fig 2H), suggesting that the structure of the mature mRNA rather than transcription plays a role in the differential localisation.

We next performed an analogous analysis to compare nuclear pre-mRNAs and nuclear polyadenylated mRNAs, estimated from NTI, and NAE, respectively (S1 Fig). We found a strong correlation between NTI and NAE (S1A Fig, $R^2$ = 0.6), suggesting that transcription rate, the common factor to both NTI and NAE ($\log_2$(NTI) = $\log_2$(Transcription rate) + $\log_2$(splicing and polyadenylation time), and similarly for NAE), has a larger effect on NTI and NAE levels than export and splicing rates. The fact that the correlation is stronger here than for $\log_2$(NAE/CAE) ($R^2$ = 0.60 vs. $R^2$ = 0.47) suggests that the variability (across all genes) in splicing times is smaller than that of cytoplasmic life times. Here, we found that $\log_2$(NTI/NAE) is more correlated with the export time (NAE) than with the splicing time (NTI) (S1 Fig, $R^2$ = 0.28 for NAE, $R^2$ = 0.015 for NTI). This observation suggests that, on average, the abundance of spliced nuclear mRNA (NAE) is more strongly determined by the variation of nuclear export time rather than splicing time of the pre-mRNA. However, transcription, splicing and export are not always independent processes and can be coupled and thus, the interpretation of the correlations may be more complicated [36–38].

## The extent of co- versus post-transcriptional splicing increases with gene length

In case of complete co-transcriptional splicing, Nuclear Total pre-mRNA (NTI) represents the population of nascent transcripts [22,39,40]. However, in Fig 1D, we observed that approximately 12% of the nuclear polyadenylated transcriptome encoding for proteins are pre-mRNA (captured in NAI samples), suggesting that a fraction of pre-mRNA is spliced post-transcriptionally. To estimate the amount of co- and post-transcriptional splicing, we compared the levels of NTI to NAI of genes belonging to the Protein Coding biotype (S2 Table). When the expression level of NAI is close to NTI, most of the nuclear pre-mRNA are fully transcribed and polyadenylated before being spliced. Conversely, a higher level of NTI compared to NAI means that nascent pre-mRNAs make up most of NTI and introns are spliced co-transcriptionally. Co-transcriptional splicing creates a typical sawtooth pattern of decreasing density of reads mapping on intron, with higher signal toward the 5′-end, described in nascent-seq datasets [23,41]. We indeed visually observed the sawtooth patterns on genes such as *Egfr* or *Zhx2* (Fig 3B), while these were absent on transcripts on the identity line (NAI = NTI), such as *Gstm6* or *Apol7a* (Fig 3C). Using the NAI/NTI ratio as proxy of co- versus post-transcriptional splicing, we observed a striking correlation of the ratio with the gene length, such that transcripts originating from a long gene sequence are more likely co-transcriptionally spliced, while short genes are more often post-transcriptionally spliced. Note that here we did not consider gene structure or the ordering of the introns. This length-dependency may reflect the situation where transcription and 3' end processing is already terminated by the time the spliceosome assembles and splices newly transcribed introns. If splicing and transcription are two processes occurring concurrently but without further dependency, the longer the gene, and particularly the longer the downstream exon, the higher the probability for the upstream intron to be detected and spliced. NTI alone, representing the total pool of pre-mRNA in the nucleus, is not influenced by the gene length (Fig 3D). The comparison of the levels of nuclear

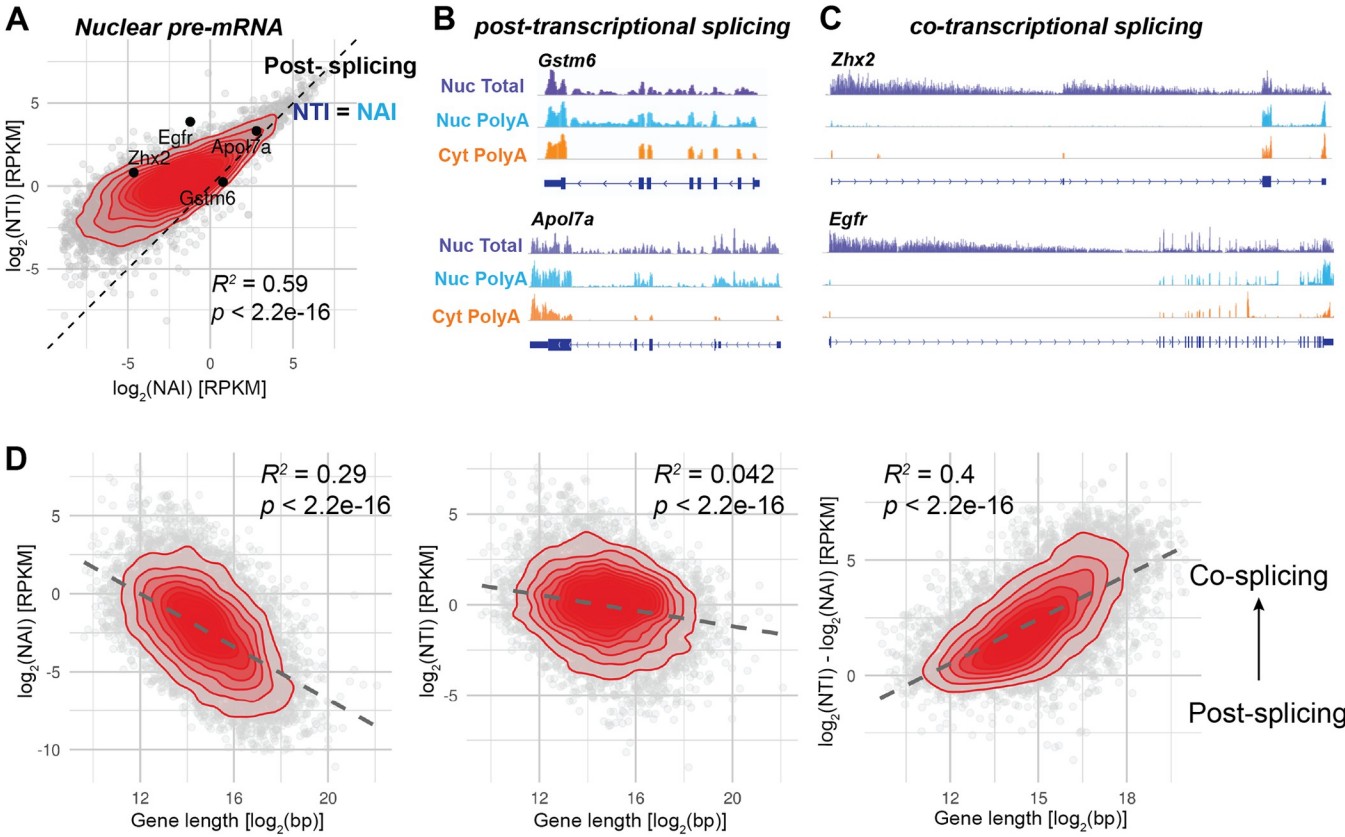

**Fig 3. The extent of co- versus post-transcriptional splicing increases with gene length. A**: Comparison of pre-mRNA expression levels in Poly(A) et Total nuclear RNA populations of protein coding transcripts (grouped by biotype): log$_2$(NTI) against log$_2$(NAI). Transcripts on the identity line (NAI = NTI, dashed line) are not spliced co-transcriptionally, while those away from the line are. Genes whose genome track are shown in B and C are shown on the plot. Red: 2D kernel density. Only protein coding transcripts are shown. **B**: Genome track view of short genes that are mainly post-transcriptionally spliced. Gene length of *Gstm6*: 4.9kb, *Apol7a*: 11kb. **C**: Genome track view of long genes that are mainly co-transcriptionally spliced. Gene length of *Zhx2*: 145kb, *Egfr*: 166 kb. Genome tracks are not to scale. **D**: Correlations of the gene length (exon + intron) with log$_2$(NAI), log$_2$(NTI) and log$_2$(NTI/NAI). The latter is related to the fraction of co- and post-transcriptional splicing. Overall, long genes are more frequently co-transcriptionally spliced than short genes.

pre-mRNA in both Total and Poly(A) conditions is therefore informative about the extent of co- vs post-transcriptional splicing, which is strongly influenced by gene length.

## Model-based identification of regulatory steps driving rhythmic cytoplasmic mRNA profiles

The accumulation of RNA transcripts in any of the subcellular compartments results from the balance between production and elimination. At steady-state (Figs 2 and S1), it was generally not possible to identify the two respective contributions. However, as we showed before, it is possible to determine these rates from time-series profiles when genes are rhythmic, using a mathematical modelling approach [12]. In essence, by modelling the profiles of causally related and oscillating RNA species, for example intronic and exonic reads from total unfractionated RNA-seq, the approach can infer production and degradation rates. We applied this method to two situations. First, a joint model of mRNA accumulation in the cytoplasm (CAE) and the nucleus (NAE) allowed us to estimate the cytoplasmic stability and assess rhythmic degradation processes (hereafter referred to as step 2, Fig 4A). Using the fractionated cell data, we now can specifically address cytoplasmic half-life instead of estimate an effective half-life that

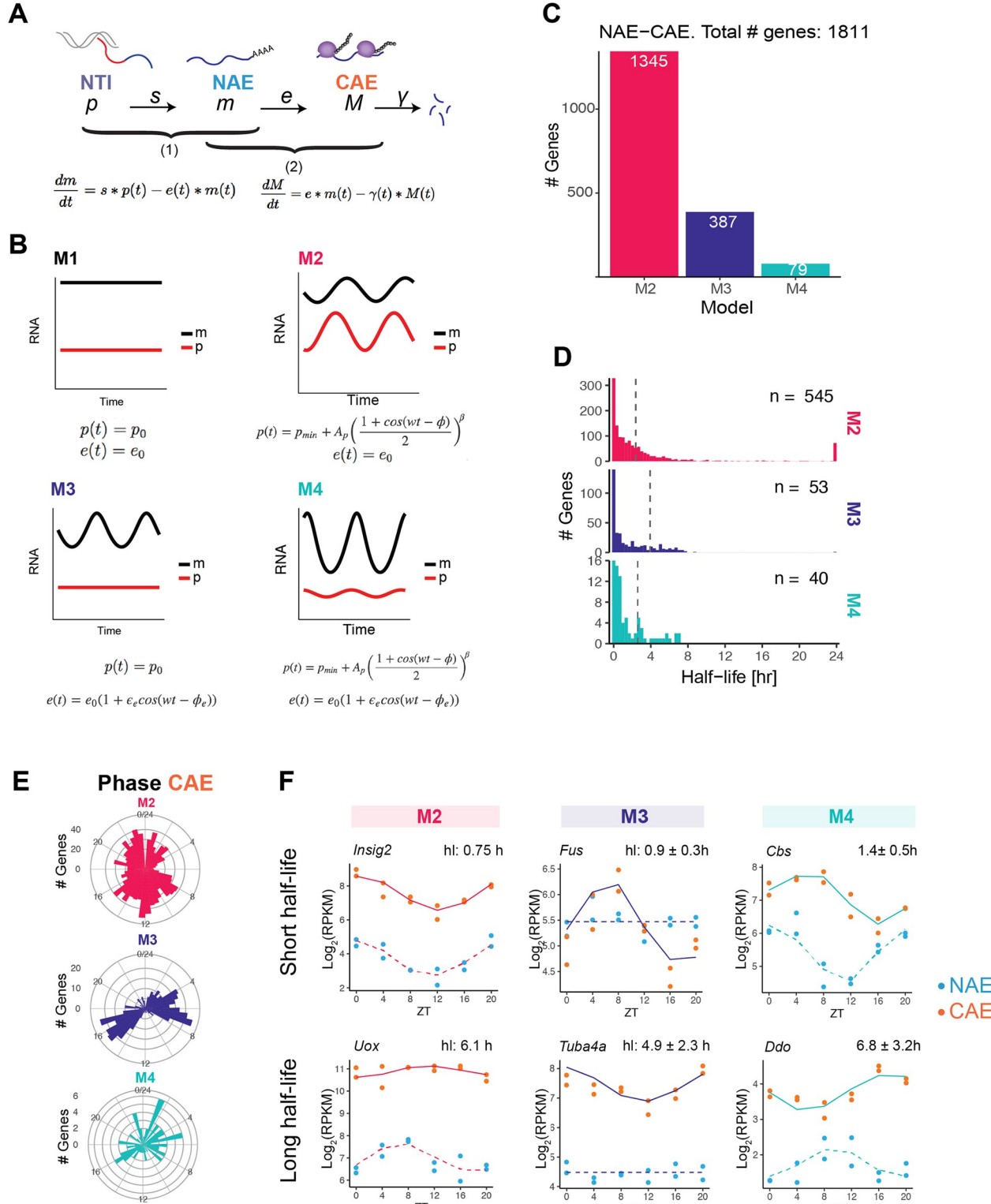

**Fig 4. Model-based identification of regulatory steps driving rhythmic cytoplasmic mRNA profiles. A**: Kinetic model describing the temporal accumulation of pre-mRNA and mRNA. RNA processing steps are split in two steps: first, we describe the temporal accumulation of nuclear mRNA (NAE) in function of splicing rate $s$ and export rate $e$ (step 1), and second, the temporal accumulation of cytoplasmic mRNA (CAE) in function of export rate $e$ and degradation rate $\gamma$ (step 2). **B**: The combination of constant or rhythmic pre-mRNA ($p$) and constant or rhythmic export in step 1 generates four models. Rhythmic pre-mRNA profile is described by a cosine function with a period of 24 hours, raised to a power β to account for

the peaked oscillatory profiles. The same equations apply to step 2, but with $\gamma$ instead of $e$, $m$ instead of $p$, and $M$ instead of $m$. **C:** Classification of genes in step 2: M2 (rhythmic NAE, constant degradation), M3 (constant NAE, rhythmic degradation), or M4 (rhythmic NAE, rhythmic degradation). **D**: Distribution of estimated cytoplasmic half-lives. To facilitate identification and considering biologically plausible ranges, we set the range of possible half-lives from 10 min to 24 h. During the optimization, $\gamma = log(2)/half\text{-}life$ sometimes reaches our fixed upper limits or lower limits. The median half-life (dotted line) is calculated excluding half-lives reaching the lower or upper boundary. **E:** Circular histogram showing the phases (peak times) of CAE in each model. **F:** Representative temporal profiles (NAE and CAE) of genes in M2, M3 and M4 genes with short half-life (up) or long half-life (bottom). Values on top of the plots indicate the half-life of the gene. In M3 and M4, mean half-lives with their corresponding min and max are indicated.

encompasses all intermediate processes occurring in the nucleus [12]. Next, we modelled the expression profiles of nuclear mRNA (NAE), which increases due to splicing of pre-mRNA (NTI) and decreases due to nuclear export (step 1). This strategy allowed us to infer the nuclear export rates and to uncover rhythmically regulated exports. Each step is described by an ordinary differential equation, where the first variable (NTI in step 1, NAE in step 2) and the rate of degradation (export $e$ in step 1, cytoplasmic degradation $\gamma$ in step 2) are either constant or rhythmic (defined by a cosine function with a phase and relative amplitude). The combination of constant or rhythmic production and degradation terms generates four different kinetic models (M1 to M4, Fig 4B). The optimal model is selected by combining a maximum-likelihood approach with the Bayesian information criterion (BIC) to control for model complexity. For genes with constant levels of pre-mRNA and mRNA (M1), only the ratio between the production and degradation can be determined. Note that this approach is different from rhythmicity assessment on one condition and has been previously applied to genotype comparisons [16]. We first applied the mathematical model to step 2 (NAE-CAE) to identify rhythmic cytoplasmic half-lives (S5 Table). Out of the approximately 11,000 genes expressed in the liver, we confidently classified 1,424 genes as having a rhythmic accumulation of nuclear RNA (M2 and M4), of which 79 are additionally rhythmically degraded (M4, Fig 4D). The rhythmic profiles of 386 genes in the cytoplasm are solely due to the temporal regulation of the cytoplasmic half-life (M3). Thus, approximately 15% of expressed genes have a rhythmic temporal profile in at least one cellular compartment. Additionally, 25% are degraded rhythmically, similar to previous studies in the mouse liver (28% in [12], 30% in [11]). The median half-lives are 2.45 h for M2, 3.88 h for M3 and 2.36 h for M4.

The distribution of peak mRNA times in the cytoplasm differs between the models. While M2 peak times were homogeneously distributed throughout the day, M3 exhibited a biphasic phase pattern with distinct associated biological functions. M3 genes peaking between ZT3 and ZT9 are involved in protein degradation such as proteasome components *Psmd3*, *Psmf1*, *Adrm1*. A second group of M3 associated genes peak between ZT14 and ZT18. This group comprises genes related to nuclear mRNA processing such as *Sfpq* and *Cpsf6*, and the splicing factors *Srsf1*, *Srsf3*, *Sfpq*, or *Thoc7*, *Hnrnpa3* and *Hnrnph1*.

In Fig 4F, we show representative genes to illustrate the typical behaviour of each model. When degradation is constant (M2), the more stable is the transcript, the more the mRNA oscillations dampen, and the longer the peak time difference between NAE and CAE. Note that in this model, this time delay cannot exceed 6 hours (it reaches 6 hours for infinitely long half-lives [25]). Transcripts with a short half-life have similar peak times (no delay) and relative amplitudes in the nucleus and in the cytoplasm (Fig 4F). Genes classified in M3 are rhythmic only in the cytoplasm. *Fus* is a well-known example of a gene rhythmically regulated at the post-transcriptional level [12,25]. Finally, genes in M4 are rhythmically regulated both at the transcription and degradation level, resulting in more complex patterns, for instance a higher relative amplitude in the cytoplasm (*Ddo*, half-life 7 h) or a longer time delay (*Cbs*, half-life 1.4 h).

## Kinetic model identifies rhythmically exported nuclear mRNAs

To uncover potentially rhythmic patterns generated by rhythmic nuclear export, we next applied the mathematical model to compare the rhythmic profiles of nuclear pre-mRNA (NTI) and nuclear mRNA (NAE) (S5 Table). Here, the export time refers to the time spent by a fully transcribed, spliced, and polyadenylated transcript in the nucleus. We confidently classified 1632 genes in one of the three models with rhythmic NAE (Fig 5A). The vast majority were rhythmically transcribed and exported with a constant rate (1428 genes as M2). 13% were exported rhythmically (M3 and M4), suggesting that nuclear export is a post-transcriptional mechanism contributing to modulation and generation of rhythmic profiles. Representative profiles of genes classified as M3, whose rhythmic nuclear mRNA accumulations were only attributed to rhythmic mRNA export, are shown in S2 Fig. Proteins encoded by these genes cover a diverse biological functions, such as deacetylation (*Lypla2*, *Sirt5*), nuclear speckles assembly and splicing (*Srrm2*, *Akap8l)*, hypermethylation of sn(o)RNA m$^7$G cap (*Tgs1*), or NAD(P)H-dependent oxidoreduction (*Rdh13*). Rhythmic export generated nuclear rhythms that propagated to the cytoplasm (S2 Fig). However, the median $\log_2$ fold-change of CAE (cytoplasmic mRNA) is lower than those originating from M2 or M4 genes (Fig 5E), but is still near 1 (2-fold amplitude). While some oscillations are damped in the cytoplasm because of the long stability of cytoplasmic mRNA, others keep oscillating with relatively large amplitudes.

The distribution of NAE peak times in M3 genes showed an enrichment around ZT8 (Fig 5C, 33% of the M3 genes peak in the time window between ZT6 and ZT10). By contrast, peak times of M2 genes, whose rhythms are generated by rhythmic transcription, exhibited the typical bimodal distribution of phases peaking at the end of the light phase (ZT22) and the dark phase (ZT10), reflecting the two waves of transcriptional activity in the liver [25,42,43]. Phases of M4 genes, whose nuclear accumulation is also regulated by rhythmic export, were distributed throughout the day. The time of shortest export time (highest export rate) was enriched in the second part of the night (half of the M4 genes had the highest export rate between ZT18 to ZT 0). The difference between the peak time of export rate and peak of transcription rate modulates the resulting pattern of NAE: rhythmic amplitudes of carbohydrate and lipid metabolism transcripts such as *Pck1*, *Lpin1* or *Ppard* are amplified by antiphasic rhythmic export, while genes such as *Sqle* or *Lipc* are damped while maintaining a phase coherence because export peak time matches transcription peak time (Fig 5D). Thus, the amplitudes of several key metabolic enzymes were boosted by a timed export process.

Nuclear residence times are known to be typically shorter compared to cytoplasmic half-lives. Indeed, previous studies estimated that nuclear lifetime ranges from 5 min to less than a couple of hours (maximal estimated nuclear residence time: 40 min in [44], 90 min in [45], 2 h in [4]). In our model of step 1, we set the range of possible export times from 5 min to 12 h. More than one third of the genes in our dataset reaches the lower boundary of 5 min (600 genes), suggesting that indeed, rhythmic transcripts are usually exported on a short time scale. The median export time, if we include those genes, is 25 min for M2 and M3 genes, and 12 min for M4 genes. Even if the majority of transcripts were processed and exported within minutes, some mRNA resided for several hours in the nucleus. For example, genes involved in steroid biosynthesis such as *Srd5a1*, *Hsd17b2*, *Dhcr7*, and *Scap*, have an estimated nuclear half-life between 2 h and 3.5 h. However, major regulators of cholesterol synthesis are exported rapidly from the nucleus (*Insig1*, *Cyp7a1*, *Hmgcr*, export time of 5 min). This suggests that despite being rhythmically transcribed, biologically related genes have distincts dynamic patterns in the nucleus. Genes involved in fructose metabolism were also enriched among long-lived transcripts (*Aldob*: 6,8 h, *Khk*: 4.8 h, *Sord*: 1.4 h, *Fbp1*: 6 h). It is noteworthy that nuclear retained

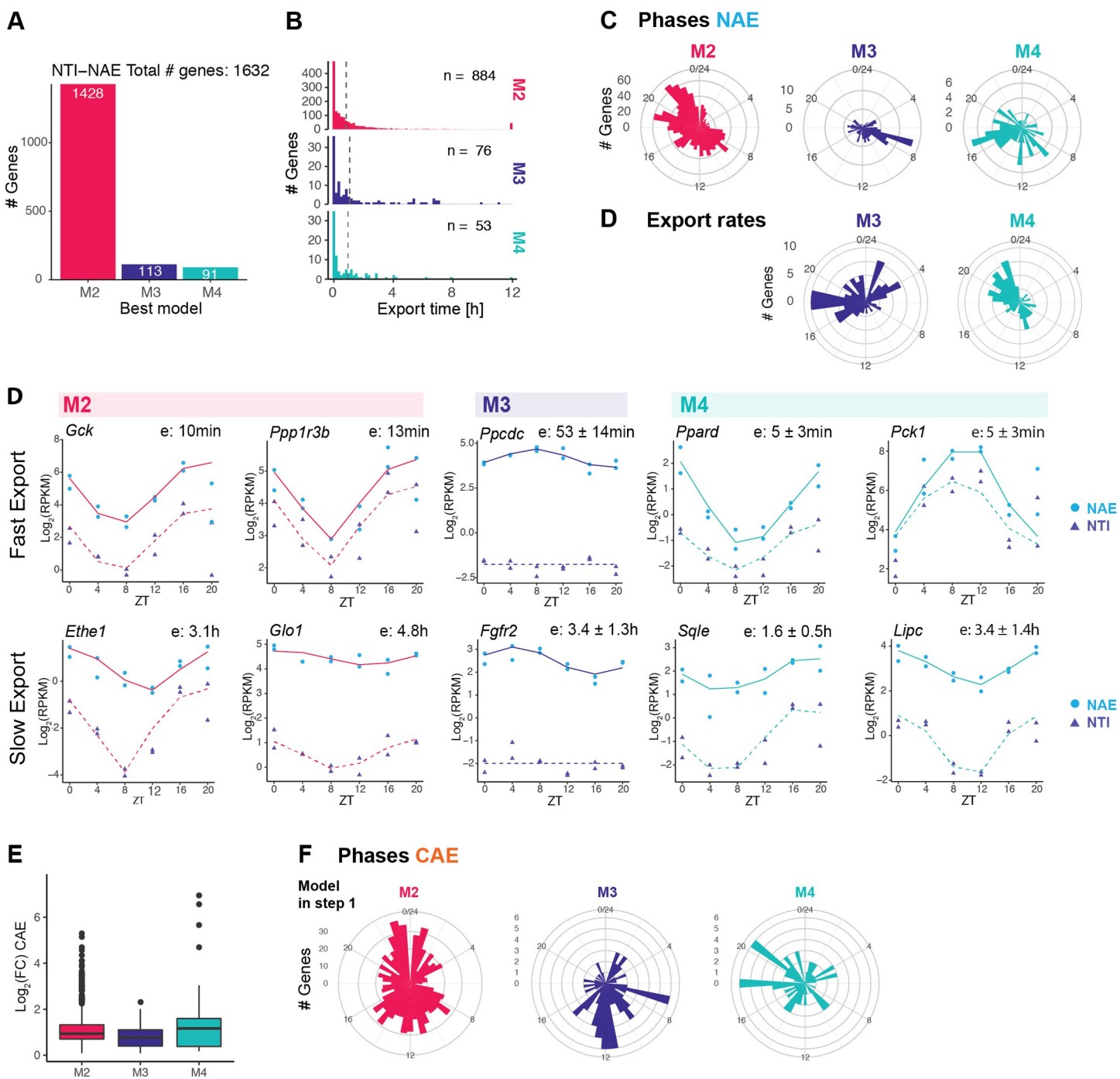

**Fig 5. Kinetic model identifies rhythmically exported nuclear mRNAs. A**: Classification of genes in M2 (rhythmic transcription, constant export), M3 genes (constant transcription, rhythmic export), or M4 (rhythmic transcription, rhythmic export). **B**: Distribution of estimated export times of the three models. The median (dotted line) is calculated excluding export times reaching the lower or upper boundary (5 min and 12 h). **C**: Distribution of peak times of rhythmic nuclear mRNA (NAE). **C**: Distribution of peak time of rhythmic export rates (shortest export time) of M3 and M4 (in M2, export is constant). **D**: Representative temporal profiles of nuclear pre-mRNA (NTI) and nuclear polyadenylated mRNA (NAE) in M2, M3 and M4 genes with short export time (up) or long export time (bottom). Additional examples of M3 genes are shown in S2 Fig. **E**: Cytoplasmic mRNA fold-change of genes classified as M2, M3 or M3 in step1. **F**: distribution of phases in the cytoplasm of genes classified as M2, M3 or M4 at step 1.

mRNAs include the glucokinase regulatory protein *Gckr* (3.2 h), genes related to thyroid hormone metabolite (*Dio1*: 2.5 h, *Thrsp*: 1.4 h) and related to glyoxylate metabolism (*Agxt*: 1.8 h, *Grhpr*: 1.9 h).

## Circadian core clock and PARbZip genes are enriched in the nucleus

Finally, we focus on the temporal behavior of core clock and PARbZip family member clock output genes. Cytoplasmic mRNA half-lives of all such genes were below 2.5 h, and nuclear half-lives ranged from 5 min to 1.1 h, allowing an efficient propagation of high-amplitude patterns (Fig 6A and 6B). Note that *Dbp* and *Rev-Erbα* (*Nr1d1*) were classified as both rhythmically exported and rhythmically degraded, which explains their particularly high relative amplitudes. As a consequence of their high cytoplasmic degradation rates, clock genes were more enriched in the nucleus compared to all cycling genes. Of interest, all three *Period* mRNAs (*Per1*, *Per2*, *Per3*) were particularly enriched in the nucleus, despite having similarly short half-lives compared to the other core clock genes, suggesting that these could be potentially slowly exported (Fig 6C). We verified by smFISH the subcellular localisation of *Per1* together with another core clock gene *Bmal1* (also named *Arntl*) on the same liver samples used for the RNA-seq (Fig 6C). Quantification of nuclear and cytoplasmic mRNA *in situ* at peak and trough expression confirmed that the nuclear-to-cytoplasmic ratio of *Per1* (1:1 at ZT12) is more nuclear than the ratio of *Bmal1* (1.6 times more cytoplasmic mRNA at ZT0). *Cry1*, another core clock gene, showed a similar nuclear-to-cytoplasmic ratio in both RNA-seq and smFISH as *Arntl* (1.5 times more cytoplasmic mRNA at ZT20, Fig 6E and 6F).

## Discussion

Gene expression is a highly regulated multistep process. We produced time series RNA measurements from fractionated liver cells to feed a simplified model, including rhythmically regulated nuclear export and cytoplasmic degradation. Cellular fractionation first allowed us to highlight characteristics and differences of the nuclear and cytoplasmic transcriptomes. Notably, at steady-state, we observed a previously reported concordance between protein and RNA localisation [30]: nuclear enriched mRNAs mainly code for nuclear regulatory proteins, and could be unstable as a way to facilitate rapid adaptation of protein production in response to external stimuli [34]. Additionally, since nuclear retention acts as a passive filter reducing transcriptional noise [45–47], retaining these transcripts could allow a stable production of proteins. On the other hand, transcripts enriched in the cytoplasm mainly code for proteins involved in house-keeping functions such as respiration and translation, presumably due to their increased stability [10,48]. Another difference between nuclear and cytoplasmic mRNAs is related to their length. Long transcripts were enriched in the nucleus, while short transcripts were more abundant in the cytoplasm. This asymmetric distribution could be again related to the cytoplasmic mRNA stability. Although our observations were made only on mouse liver tissue, a link between transcript length and decay rate was hinted at in Drosophila Kc167 cells [6]. Since mRNA degradation that occurs through the deadenylation of the poly(A) tail and a subsequent exonuclease attack [34] has not been linked with transcript length, the higher instability of long transcripts may reflect an increased probability to be stochastically attacked by endonucleases [6].

Another process that depends on genomic length is splicing. Splicing mainly occurs co-transcriptionally in most organisms [41,49–51]. However, the proportion of introns that are removed concurrently with transcription is still under debate, and is highly variable depending on the organism and experimental methods [52]. For example, pre-mRNAs in mouse liver are

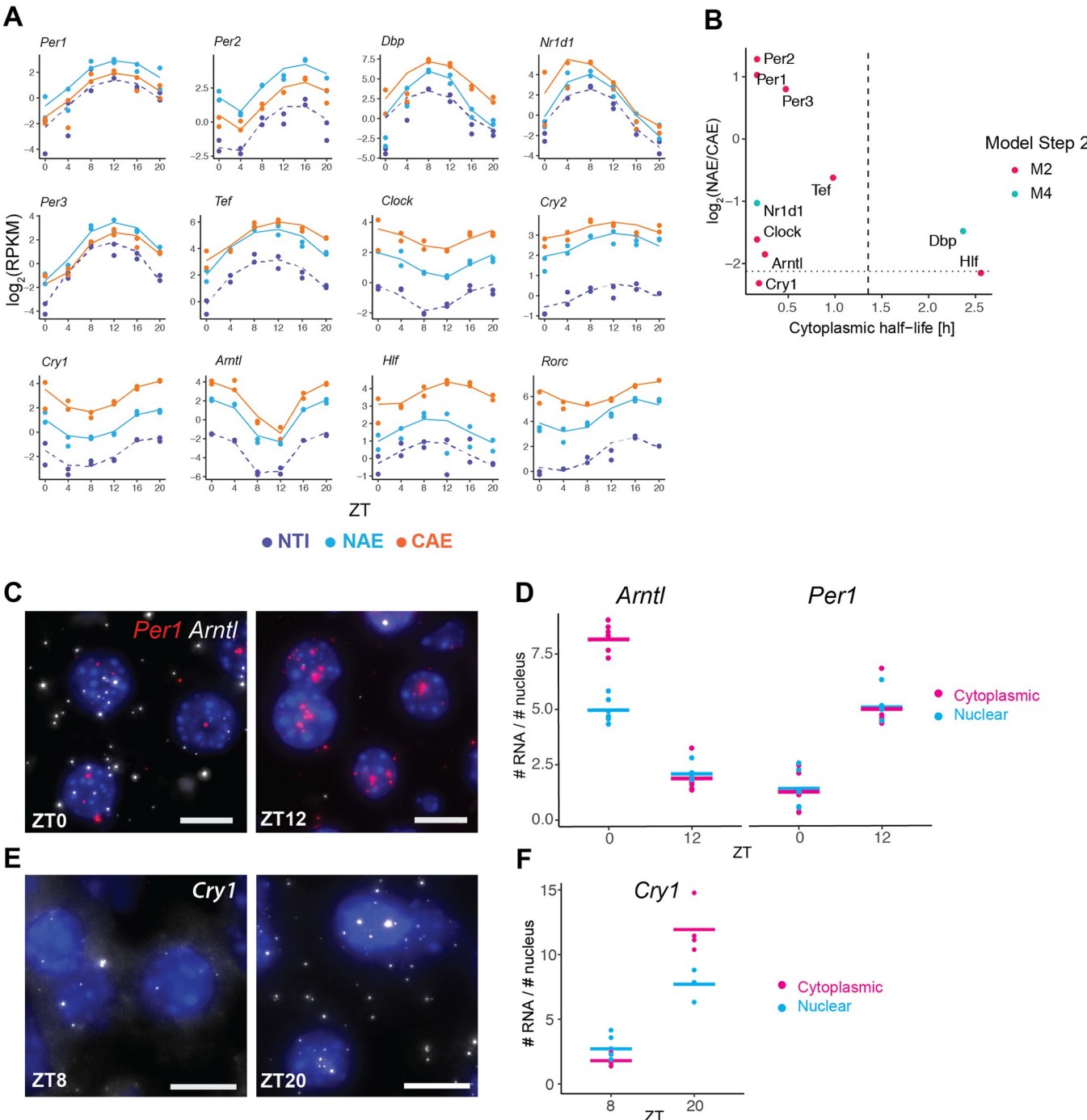

**Fig 6. Circadian core clock and PARbZip genes. A**: RNA-seq temporal profiles of clock genes and clock-controlled genes (PARbZip family members), sorted by their nuclear-to-cytoplasmic ratio. **B:** log₂(NAE/CAE) versus cytoplasmic half-life estimated at step 2. Dashed vertical line represents the median half-life times of all rhythmic mRNAs. Dotted horizontal line represents the median log₂(NAE/CAE) ratio of all rhythmic mRNAs. **C** and **E:** smFISH of core clock mRNAs *Per1*, *Arntl*, and *Cry1* at their maximal and minimal expression levels in mouse liver tissue. Nuclei stained with DAPI (blue). Scale bar: 10μm. 63x magnification. **D** and **F:** Quantification of nuclear and cytoplasmic mRNA transcripts on smFISH images. The nuclear-to-cytoplasmic ratio of *Per1* is significantly more nuclear than *Arntl* and *Cry1* at their respective peak time (pairwise t-test, Bonferroni correction, p-value $< 10^{-5}$) and $< 10^{-4}$ respectively). 4–6 images were taken per animal, with each field of view containing on average 500 nuclei. The average number of mRNA molecules per number of nuclei in each image is represented (dots). Number of nuclei analysed: *Arntl* and *Per1*: 3237 (ZT0), 3300 (ZT12). *Cry1*: 1714 (ZT8) and 1319 (ZT20).

less frequently spliced co-transcriptionally compared to Drosophila [53]. We investigated the proportion of co- versus post-transcriptionally spliced genes by comparing the expression level of intronic signals in nuclear total pre-mRNA (NTI) and the nuclear polyadenylated pre-mRNA (NAI). We noticed a strong correlation with gene length, namely, long pre-mRNA showed more frequent co-transcriptional splicing compared to shorter pre-mRNAs. Similar observations on the dependency of gene length and splicing regime were found in a previously published dataset in mouse embryonic stem cells, where longer introns were more likely removed post-transcriptionally [54]. Our observations thus support that when a gene is short, transcription is already completed by the time the intron is recognised and spliced. If the gene is long, the spliceosome has more time to recognise splice sites, assemble on the pre-mRNA, and cleave the intron. This relationship with gene length was stronger when focussing on protein coding transcripts only, which could reflect the different processing mechanisms regulating distinct classes of RNAs [1]. It is also known, however, that transcription and splicing can be coupled through physical interactions of the associated protein machineries, which can also be favoured when these factors are concentrated in one location, such as in nuclear speckles and paraspeckles [55]. Such coordination of transcription and splicing processes also modulates alternative splicing events [56], which was not specifically assessed in this work. Indeed, for simplicity and due to technical limitations of obtaining sufficiently accurate isoform-specific expression levels, we have summarised our results at the gene level by weighting the isoforms by their estimated expression levels. We also did not consider variations in splicing efficiencies of introns within a gene body, which differ depending on the length of the downstream exon or on whether flanking exons are alternatively or constitutively spliced [3,51,57]. In the future, more sophisticated methods such as long-read sequencing would be necessary to refine the splicing models [58]. Indeed, while being highly promising, long-read sequencing technologies are currently limited in the length of sequenced transcripts (15kb to 30 kb) and are still biased toward short length transcripts [59].

We next investigated the temporal dimension of our dataset to identify possible new layers of circadian gene regulation, in addition to the many mechanisms mentioned in the introduction, notably chromatin states, transcription, alternative splicing, translation, protein accumulation and localization (reviewed in [60]). We adapted our previously developed models that take advantage of the relationship between relative peak times and relative amplitudes describing the propagation of oscillations to estimate the cytoplasmic degradation rates and nuclear export rates of approximately 1400 genes. These differential equations models describe the lifecycle of an RNA transcript, for example the nuclear pre-mRNA (NTI) is transformed into mRNA (NAE) through an effective splicing rate that combines the splicing rate of the individual introns, and which is then exported at a given rate. These are clearly highly simplified views of the complex processes, and it also remains challenging to infer those parameters from the data fits. Notably, the splicing time is identifiable only for some of the models where we have set a prior on it, in other models (e.g. static) only a relative splicing time can be obtained (Methods). Further experimental validations would be needed to assess the accuracy of the predicted splicing times. With those models, we found a median cytoplasmic half-life of 2.5 h for cycling transcripts. In general, mRNA half-lives range from minutes to several hours, and the estimation varies depending on the model (cell type, tissue) and method (total RNA-seq, transcription inhibition, metabolic labelling). As a comparison, median half-lives of mRNA was 9 h in NIH3T3 cells [8], 3.9 h in mouse ESCs [61], 4 h in HEK293 cells [62] 3.4 h in HELA cells [48], and 2.1 h in another study with NIH3T3 fibroblasts [63]. Our shorter estimations can be explained by our focus on rhythmic genes, which need to be relatively short-lived to maintain oscillations. We also estimated that 25% of mRNA in the cytoplasm are degraded cyclically, which offers flexibility to boost amplitudes and fine-tune phases [11, 12, 25, 64].

Such fluctuations of mRNA stability can occur through the activity of plethora of RNA-binding proteins [65–67], through silencing by miRNA [68], by regulating poly(A) tail length [69], and regulating deadenylase activity [70].

Nuclear export times are overall shorter and the majority of the rhythmic nuclear mRNA have a half-life of less than 25 min. Our experimental design reliably detects long export times on the order of our sampling time (in the range of hours), but does not allow a precise quantification of processes happening at extremely short time-scales. For a majority of genes, ignoring the nuclear export step as we and others previously did [11,12] is a valid assumption due to its small contribution to the total mRNA lifetime. However, when nuclear lifetime significantly contributes to the total RNA lifetime, our dataset allows us to separate the processes and provide a more specific interpretation of the kinetic processes. In addition, we were able to uncover rhythmic patterns in the nucleus, which strongly suggests that the nuclear lifetime varies around the day for 12.5% of the rhythmic nuclear mRNA. As a result, amplitudes of several rhythmically exported mRNA such as *Pck1*, *Por* and *Lpin1* involved in liver metabolism were boosted. Rhythmic export could be modulated by nuclear RNA processing factors. For example, the components of nuclear paraspeckles NONO and SFPQ, display rhythmic nuclear accumulation profiles in mouse liver cells [26]. Interestingly, NONO binds around a thousand mRNA in the liver, of which about one third exhibits circadian accumulation [29]. The deletion of NONO in a knock-out mouse model altered the rhythmic profiles of hundreds of mRNA profiles (but not systematically of the pre-mRNAs) by advancing or delaying the phase, or abolishing the rhythm, suggesting a pleiotropic role of NONO in the post-transcriptional regulation of rhythmic nuclear mRNA accumulation. The role of paraspeckles in regulating rhythmic export was established by transfecting a reporter mRNA with an IRAlu element fused in its 3'UTR in rat pituitary cell line, which was retained in the nucleus and released in the cytoplasm in the circadian manner [71]. Thus, nuclear export is most likely one of the many regulatory layers of rhythmic gene expression in mammalian cells.

The temporal analysis of our dataset was performed only at the whole gene-level and we therefore did not look into differential rhythmicity of splice isoforms. In previous works on circadian alternative splicing, it was found that mRNAs encoding for splicing-related genes frequently exhibited 24-h rhythmic expression with a bimodal distribution of peak phases across tissues and species, indicating that they might be controlled by the circadian clock [72,73]. Concomitantly, rhythmic alternative splicing was found to be widespread across mammalian tissues and conserved alternative splice isoforms that peaked at opposite times of the day were reported in oncogenes and tumour suppressors [73]. In the liver, it was reported that the feeding/fasting cycle modulates hepatic circadian alternative splicing in an exon-dependent manner [74]. In general, there is a risk that some circadian regulation will be masked with gene-level analyses. Nevertheless, inference of alternative transcript isoforms remains challenging with short read sequencing [59], and stratifying the analysis of the half-lives by transcript isoforms would make our dynamical models too complex. Advances in long-read sequencing technologies would allow refined isoform-levels dynamic transcriptome analysis, contributing to a better understanding in physiological or pathological conditions [73].

In this work, we provided a comprehensive and temporal inventory of RNA nuclear and cytoplasmic localisation in mouse livers, as well as an identification of the nuclear or cytoplasmic origins of transcript rhythms. In particular, this analysis suggests that mRNA oscillations can be post-transcriptionally modulated also at the level of nuclear export, and contributes to a better understanding of the dynamic regulation of the transcriptome over the 24 hours day.

## Methods

### Ethics statement

All animal care and handling were approved by the Canton de Vaud laws for animal protection (authorisation VD.3197).

### Animal housing and mouse liver sampling

All mice experiments were performed on male WT mice (C57/BL6J) aged from 8 to 12 weeks old, and housed in 12:12 light:dark cycle. 3 days prior to sacrifice and sampling day, mice were given access to food only during the active phase (night-restricted feeding). Two mice were sacrificed by decapitation every 4 hours (ZT0, 4, 8, 12, 16 and 20). ZT0 is the beginning of the light phase, and ZT12 is the beginning of the dark phase. Pieces of liver were immediately collected after sacrifice for RNA extraction and histological analysis (single-molecule RNA-FISH).

### Cellular fractionation and RNA extraction

Nuclear and cytoplasmic liver mRNA were isolated according to the Nascent-seq protocol [23] with few modifications. Briefly, approximately 300mg of the big lobe were homogenised with a Potter-elvehjem PTFE pestle (DWK Life Sciences #9651693, Aesch, Switzerland) together with 3.5ml of ice-cold PBS (Thermo Fisher Scientific #AM9625, Waltham, USA) and 3.5ml of homogenisation buffer (2.2M sucrose (Sigma Aldrich #84097, St. Louis, USA)), 15mM KCl (Sigma-Aldrich, #P9541), 2mM EDTA (Thermo Fisher Scientific #AM9261, 10mM Hepes pH 7.6 (Sigma-Aldrich #J61047), 0.15 mM spermine (Sigma-Aldrich #S1141), 0.5mM spermidine (Sigma-Aldrich #S2626), 0.5 mM PMSF (Sigma-Aldrich #329-98-6), 1mM DTT (Thermo Fisher Scientific #R0861), 20 µg/ml aprotinin (AppliChem #A2132, Darmstadt, Germany), 2.5µg/ml leupeptin (Sigma-Aldrich #L2884), 100µg/ml pepstatin (AppliChem #A2205), and 10U/ml RNAseOUT RNase inhibitor (Invitrogen #10777019, Carlsbad, USA)). The homogenate was mixed with the remaining homogenisation buffer (total volume of homogenisation buffer: 18ml) and layered on top of 7.2ml of a sucrose cushion buffer (2.05 M sucrose, 10% glycerol, 15mM KCl, 2mM EDTA, 10mM Hepes pH 7.6, 0.15 mM spermine, 0.5 mM spermidine, 20 µg/ml aprotinin, 2.5 µg/ml leupeptin, 100 µg/ml pepstatin, 0.5 mM PMSF, 0.1 M DTT, 10 U/ml RNAseOUT RNase inhibitor). The liver homogenate was ultra-centrifuged in a SW28 rotor (Beckman Coulter #342207, Brea, USA) at a speed of 103'000 $g$, at 4˚ for 1 hour. Supernatant (cytoplasmic fraction) was mixed with ethanol (final ethanol concentration of 70%) and stored overnight at -80˚C. Cytoplasmic extracts in ethanol were centrifuged at 5000 rpm for 15 min at 4˚C to precipitate RNA. Ethanol was discarded, and the pellet was resuspended in 800ul of QIAzol Lysis reagent (QIAGEN #79306, Venlo, Netherlands). RNA was extracted on spin columns using the miRNeasy Kit (QIAGEN #74104) according to the manufacturer's instructions. After ultracentrifugation, the pellet (nuclei) was resuspended in 3ml of Nuclear RNA extraction buffer (guanidium thiocyanate 50% (w/v) (Sigma-Aldrich #G9277), sodium citrate 0.75M pH7 (Sigma-Aldrich #1613859), 200µM sodium acetate 2M pH 4 (Sigma-Aldrich #71180) and 1% β-mercaptoethanol (Sigma-Aldrich #M6250)), and was mechanically homogenized through a syringe needle 26G. 3ml of water-saturated phenol pH 4.5 (Fisher Scientific #802517) and 1200µl of chloroform:isoamyl alcohol 24:1 (Sigma-Aldrich #25666) were added to the resuspended pellet, and were well shaken before resting for 15 min at room temperature. The mix was centrifuged at 4˚C at 1500$g$ for 20 min. The supernatant was resuspended in ethanol (final volume 60% (v/v)). RNA is then extracted on spin columns using miRNEasy kit according to manufacturer's instructions, including a DNAse treatment (QIAGEN #79254, Venlo, Netherlands).

For total liver RNA extraction, pieces from the main liver lobes were frozen in liquid nitrogen. About 20mg were frozen and used for RNA extraction using the miRNEAsy kit.

## RNA-sequencing

To sequence polyadenylated nuclear and cytoplasmic RNA, we used the TruSeq stranded mRNA LT kit (Illumina #20020594, San Diego, USA), starting from 650 ng of RNA, according to the manufacturer's protocol. To sequence total RNA (with ribosomal RNA depletion), we used the KAPA RNA HyperPrep Kit (Kapa Biosystems #KK8540, Wilmington, USA) with RiboErase prep (Kapa Biosystems #KK8560, Wilmington, USA) combined with KAPA Unique Dual-Indexed Adapter Kit Illumina Platforms (Kapa Biosystems #KR1736, Wilmington, USA), starting from 650 ng RNA. Libraries were sequenced on a Hiseq4000 (Illumina, San Diego, USA). We used a SBS 50 cycles and SR cluster kit, performing a single read sequencing of 65 nucleotides. We sequenced on average 70M reads for Nuclear PolyA and Nuclear Total samples, 25M reads for Cytoplasmic PolyA and Cytoplasmic Total samples, and 50M reads for the Unfractionated samples.

Quantification of pre-mRNA and mRNA expression levels was performed using Kallisto version 0.46.0 [28]. A combined index was built using reference fasta files from Ensembl for *mus musculus* (GRCm38). In particular, pre-mRNA (introns and exons) and mRNA (exons) reference sequences were used as an input and named accordingly. Additionally, we had added spike-in mRNA originating from rat liver (male Wistar, donated by the Laboratory for Functional and Metabolic Imaging, EPFL) accounting for 5% of the total mass liver, but these were in the end not used. Therefore, the reference genome used for the mapping includes fasta files from *rattus norvegicus* (Rnor_6.0). Pseudoalignment and quantification were run on the aforementioned combined Kallisto index with parameters "—single—rf-stranded -l 100 -s 30 -s 30". TPM (transcript per million) and estimated counts at (pre-) transcript level were used for further analysis.

We filtered lowly expressed isoforms based on the expression level of the mature mRNA. Specifically, we calculated the Isoform Fraction (IF) for each transcript, which is defined as the TPM (transcript per million) of the isoform divided by the sum of TPM of all isoforms of a given gene. We defined a gene-specific threshold that depends on the number of annotated isoforms (n): the IF had to be greater than $1/n$ to be kept. Additionally, we set the maximal threshold to 0.2, such that when a gene has only two isoforms, if one contributes to more than 20% of the total expression, it was not filtered out. If a transcript met the condition in at least 6 samples out of the 60 samples (5 RNA populations, 12 time-points), it is kept.

In addition to the isoform-specific quantification, we pooled for each gene the isoforms belonging to the same biotype in order to have a biotype-specific quantification for each gene (sum of the estimated counts of the isoform belonging to the same biotype). As for the gene length used for normalisation, we calculated the weighted effective gene length (estimated by Kallisto), where weights were the expression levels of each isoform in each condition (therefore, gene length may slightly differ between RNA populations). We created 8 biotypes, based on the Ensembl description. Protein coding (PC), Retained Intron (RI), snRNA, and Processed Transcripts are defined by Ensembl. Nonsense mediated decay and non stop decay are grouped as "NMD". We grouped lincRNA, bidirectional promoter lncRNA, sense overlapping, sense intronic, 3prime overlapping ncRNA, lncRNA and in one biotype named "long non-coding RNA" (lncRNA). Small Nucleolar RNA (snoRNA) comprises the biotypes "snoRNA" and "scaRNA". All the biotypes containing the keyword "pseudogenes", whether they are processed, transcribed or not are grouped as "Pseudogene".

Finally, we also summed all the transcripts by gene, independently of their biotype, in order to obtain gene-wise quantification. The gene length was calculated as the weighted average based on the relative expression of all the isoforms in each RNA population.

For the analysis at steady-state (Figs 1–3 and S2 and S3 Tables), we normalised the estimated counts by Kallisto (S1 Table) into RPKM using the R package "DESeq2" [75]. This normalisation procedure was applied on each RNA population separately. We used the combined library size (pre-mRNA + mRNA) per sample to adjust for sequencing depth. The size factor was estimated using mRNA only, such that the median level of mRNA is aligned for all the samples. The same size factor was applied to the pre-mRNA, such that the relationship between pre-mRNA and mRNA is conserved. Here, we directly implemented a rhythmic analysis (harmonic regression) during the DESeq2 normalisation process, using the following model:

$$\log_2(\mu_{g,s,t}) = a_{g,s} * \cos(w * t_s) + b_{g,s} * \sin(w * t_s) + \text{intercept}$$

where $\mu_{g,s,t}$ is the mean count of the negative binomial distribution for gene $g$ for sample $s$ at circadian time-point (ZT) $t_s$. $a_{g,s}$ and $b_{g,s}$ are the coefficients of the cosine and sine functions, and $w = 2 * \pi / 24$ h. The phase (peak of expression) is defined as $\tan^{-1}(b/a)$ and the amplitude is (or $\log_2$ fold change) as $a^2 + b^2$. The full model with rhythmic parameters is tested with a Likelihood Ratio test against the reduced model with an intercept only. The intercept fitted by DESeq2 is used as the mean expression level over the 6 time points.

Finally, to express these expression levels in absolute mRNA counts per hepatocyte, we used the data in [4], where the authors converted the counts obtained from sequencing nuclear and cytoplasmic PolyA RNA from mouse liver into number of molecules per cell using smFISH. We applied a linear regression model on the $\log_2$-transformed RPKM and added the following $\log_2$ offsets to the NA, NT, CA and CT datasets: -1.34, -0.15, +1.2 and +0.7.

## Modeling temporal RNA-seq profiles

To describe the accumulation of each RNA species in the nucleus and in the cytoplasm, we assumed that the pre-mRNA $p$ is spliced and polyadenylated at a rate of $s$ to produce mature nuclear RNA $m$. Then, $m$ is exported at a rate $e$ into the cytoplasm and becomes $M$, where it is finally degraded at a rate $\gamma$. We used NTI to approximate the level of pre-mRNA $p$, NAE and CAE to describe the accumulation of mature RNA in the nucleus $m$ and in the cytoplasm $M$. We modeled the accumulations of $m$ as a function of $p$, and of $M$ as a function of $m$, with two separate ordinary differential equations:

$$\frac{dm(t)}{dt} = sp(t) - e(t)m(t) \tag{1}$$

$$\frac{dM(t)}{dt} = em(t) - \gamma(t)M(t) \tag{2}$$

To distinguish different modes of post-transcriptional regulation, we applied the model-based approach described in [12] on the two equations Briefly, $e(t)$ and $\gamma(t)$, respectively the export and cytoplasmic degradation rates, can be either constant ($e(t) = e_0$ and $\gamma(t) = \gamma_0$) or rhythmic ($e(t) = e_0(1 + \varepsilon_e \cos(wt - \varphi_e))$ and $\gamma(t) = \gamma_0(1 + \varepsilon_\gamma \cos(wt - \varphi_\gamma))$), with w the angular frequency of $2\pi/24h^{-1}$, $\varphi$ the phase (peak time) and $\varepsilon$ the relative amplitude varying from 0 to 1. $p(t)$ and $m(t)$ are either constant ($p(t) = p_0$ and $m(t) = m_0$) or rhythmic ($p(t) = p_{min} + A_p((1 + \cos(wt - \varphi_p))/2)^\beta$ and $m(t) = m_{min} + A_m((1 + \cos(wt - \varphi_m)/2)^\beta$, with A the absolute amplitude (peak to trough), $\beta$ ranging from 1 to 2, and $\varphi$ the phase. Because NTI, NAE, and CAE come from samples processed and sequenced separately, the relative abundance is not defined, therefore, these parameters do not have a biologically interpretable value.

The following procedures and equations describe step 1, but are the same for step 2. To infer the values of the 8 parameters: $A_p$, $Min_p$, $\varphi_p$, $\beta$, $s$, $e_0$, $\varepsilon_e$ and $\varphi_e$, we used the function *optim*

from the R package "stats" with the method "L-BFGS-B". We assumed that RNA-seq counts data follows a binomial distribution [75]. Thus, the log-likelihood function to be minimised for each gene is:

$$log(L) = \sum_t logNB((n_p(t)|\mu_p(t), a_p(t)) + logNB(n_m(t)|\mu_m(t), a_m(t))$$

with

$$\mu_p(t) = p(t)S(t)L_p$$

$$\mu_m(t) = m(t)S(t)L_m$$

Here, $n(t)$ is the read count, the subscripts $p$ and $m$ stand for pre-mRNA (NTI) and nuclear mRNA (NAE). α is the dispersion parameter of the negative binomial distribution, which is specific to each gene and time-point. It is estimated by the R package DESeq2 (fit type: "parametric") [75]. $\mu(t)$ is the expected mean of counts, which corresponds to the concentration of transcripts ($p(t)$ and $m(t)$) multiplied by the gene length, and a sample-specific scaling factor $S$, and further multiplied by $10^9$ to obtain numerically convenient values. The scaling factor $S$ is first estimated by DESeq2 separately on each RNA population, including only mRNA, such that the median expression levels of mRNA are aligned between samples (Nuclear Total, Nuclear PolyA, and Cytoplasmic PolyA). The parameter $e$ is bounded to 5 min to 12 h, and $\gamma$ to 10 min to 24 h.

The combination of rhythmic / constant transcription and export terms generates 4 models (M1 to M4), each of them are fitted to experimental RNA-seq profiles and the most parsimonious model was chosen based on the Bayesian information criterion (BIC). The BIC is defined as:$-log(L) + Klog(N)$, with $K$ the number of parameters and $N$ the number of datapoints. In order to discuss only unambiguously classified genes, we selected genes with a BIC probability (Schwarz weights) $> 0.6$.

## Gene enrichment analysis

We defined genes that were nuclear and cytoplasmic enrichments using DESeq2 [75]: raw counts of NAE and CAE samples were tested for enrichment using the cellular localisation (Nuclear or Cytoplasmic) as a variable in the design formula (S4 Table). Contrary to the design used for S1 Table, time was not used as a variable in the formula. Genes with a $log_2$FC $>2$ and an adjusted p-value $<0.01$ were considered as enriched in one of the compartments, which represents the top 15% most nuclear and most cytoplasmic genes. We tested for gene set enrichment of protein coding genes in nuclear and cytoplasmic fractions using Gene Ontology Terms derived from "Biological Processes". We tested the significance of the enrichment using Fisher's exact test implemented in the EnrichR package [76] (version 3.0).

## Single-molecule RNA-FISH

Dissected liver pieces were immediately fixed in 10% Neutral buffered Formalin (Sigma-Aldrich #HT5012) at 4C° for 24 to 36 hours. Fixed samples were then washed in PBS 1x for 30 min, and dehydrated in standard ethanol series followed by xylene bath, and finally embedded in paraffin (formalin fixed paraffin embedded, FFPE tissues). Sections were 8μm thick. Single-molecule RNA Fluorescent *in situ* Hybridisation (smFISH) experiments were performed using the RNAScope Multiplex Fluorescent Assay V2 (Advanced Cell Diagnostics #323100, Newark, USA), according to manufacturer's instructions. Nuclei were counterstained with DRAQ fluorescent probe solution (Thermo Fisher Scientific #62251) for 10 min. Sections were mounted

with ProLong Gold Antifade Mountant (Invitrogen #P36930). The *Cry1* smFISH images were taken from [27] from liver cryosection and quantified for nuclear and cytoplasmic abundance. Catalogue number of the probes: *Cry1*: 500031, *Per1*: 438751, *Arntl*: 438741, *Actb*: 316741, *Mlxipl*: 558141.

## Microscopy and image analysis

Sections were imaged with the Visitron Spinning Disk CSU W1 with a motorised stage. Z-stacks were acquired with a distance of 0.2μm between each Z position (~40 images per frame) with an oil immersion 63x objective. All z-stacks were maximally projected for analysis. The image analysis was performed using ImageJ ([77] software version 1.53c, Java 1.8.0_66). mRNA spots are detected by first applying a Gaussian blur, followed by the edge-detector Laplacian filter (FeatureJ). The local maxima—corresponding to the spots—are computed and counted. Nuclei are detected by applying a median blur filter, the Otsu method for automatic thresholding, and the watershed algorithm for segmentation. We quantified a density of mRNA per nuclei: we counted all the mRNAs present in one microscopy image and divided by the number of segmented nuclei.

## Supporting information

**S1 Fig. The ratio of nuclear pre-mRNA and mRNA reflects the relationship between transcription rate, splicing time, and export time. A**: NTI versus NAE, in $\log_2$(RPKM), averaged over 6 time points. According to our model, $\log_2$(NTI) is defined as $\log_2$(Transcription rate) + $\log_2$(pre-mRNA processing time (splicing and polyadenylation)) and $\log_2$(NAE) is defined as $\log_2$(Transcription rate) + $\log_2$(export time). In grey: kernel 2D density. Boxplots on top and on the right show the distribution of NTI and NAE. **B**: NTI / NAE ratio in $\log_2$-scale, defined as $\log_2$(splicing time)—$\log_2$(export time), against the average expression of NTI (up) or against NAE (bottom). Grey line indicates linear regression.
(TIF)

**S2 Fig. Temporal profiles of genes classified as M3 in step 1.** 25 profiles of genes classified as M3 (constant transcription, rhythmic export) according to the following criteria: all the genes have a $\log_2$FC > 0.8, and all classified as M2 or M4 in the analysis of step 2 (NAE vs CAE). Plots are arranged by the expression level of NAE (highest NAE *Srrm2* in the upper left corner). Light blue dots and solid line: NAE, purple dots and dotted line: NTI.
(TIF)

**S1 Table. RNA-seq counts of ribosome-depleted total RNA of nuclear, cytoplasmic, and unfractionated liver tissue, and polyadenylated RNA of nuclear and cytoplasmic liver tissue.**
(XLSX)

**S2 Table. Normalised RNA-seq counts of ribosome-depleted total RNA of nuclear, cytoplasmic, and unfractionated liver tissue, and polyadenylated RNA of nuclear and cytoplasmic liver tissue.**
(XLSX)

**S3 Table. Parameters of the linear harmonic regression on normalised RNA-seq counts of ribosome-depleted total RNA of nuclear, cytoplasmic, and unfractionated liver tissue, and polyadenylated RNA of nuclear and cytoplasmic liver tissue.**
(XLSX)

**S4 Table. GO Terms analysis of polyadenylated nuclear and cytoplasmic polyadenylated RNA.**
(XLSX)

**S5 Table. Estimated kinetic parameters and selected models at step 1 and step 2.**
(XLSX)

## Author Contributions

**Conceptualization:** Clémence Hurni, Benjamin D. Weger, Felix Naef.

**Data curation:** Clémence Hurni, Cédric Gobet.

**Formal analysis:** Clémence Hurni, Benjamin D. Weger, Cédric Gobet.

**Funding acquisition:** Felix Naef.

**Investigation:** Clémence Hurni, Benjamin D. Weger.

**Supervision:** Felix Naef.

**Visualization:** Clémence Hurni.

**Writing – original draft:** Clémence Hurni.

**Writing – review & editing:** Clémence Hurni, Benjamin D. Weger, Cédric Gobet, Felix Naef.

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
