## [Decision Letter · Decision Letter 0]

8 Dec 2021

Dear Dr Naef,

Thank you very much for submitting your Research Article entitled 'Comprehensive analysis of the circadian nuclear and cytoplasmic transcriptome in mouse liver' to PLOS Genetics.

The manuscript was fully evaluated at the editorial level and by independent peer reviewers. The reviewers appreciated the attention to an important problem, but raised some substantial concerns about the current manuscript. Based on the reviews, we will not be able to accept this version of the manuscript, but we would be willing to review a much-revised version. We cannot, of course, promise publication at that time.

If you decide to revise the manuscript for further consideration at PLOS Genetics, please aim to resubmit within the next 60 days, unless it will take extra time to address the concerns of the reviewers, in which case we would appreciate an expected resubmission date by email to plosgenetics@plos.org.

[LINK]

We are sorry that we cannot be more positive about your manuscript at this stage. Please do not hesitate to contact us if you have any concerns or questions.

Yours sincerely,

Achim Kramer

Associate Editor

PLOS Genetics

Gregory Barsh

Editor-in-Chief

PLOS Genetics

Reviewer's Responses to Questions

**Comments to the Authors:**

Reviewer #1: Many genes oscillate in a time of the day dependent manner – being regulated either directly through the cells circadian clock or through systemic signals such as body temperature cycles. While transcriptional regulation is classically considered the main source of oscillating gene expression, an important contribution of post-transcriptional mechanisms is likely. In the present manuscript the authors use RNAseq data of circadian liver samples thereby comparing nuclear and cytoplasmic PolyA+ and Ribo- libraries. The combination of these libraries allows the estimation of nuclear export and cytoplasmic degradation rates, and reveals many oscillating genes with rhythmic changes in these parameters. Overall the study is well designed and includes important control experiments.

I have a few minor points:

1) Page #5: The text says PC2 accounts for 17% of the variability (Figure: 14.7)

2) The authors sequenced polyA+ and Ribo- also of the cytoplasmic RNA. Are there considerable differences between rhythmic genes found in polyA+ vs Ribo- for the cytoplasm?

3) The PCA blot in Figure 1C shows a very nice clustering of the different libraries. Does this contain all timepoints and could the timepoints be labeled?

4) Page #9: Is Gck with a negative log2 of -1.2 nuclear enriched? What is the cut-off?

5) In the text the authors use r values while the figures depict R² values. Could they decide for one?

6) All linear regression fits show an identical p Value of < 2.2e-16. Even if almost no correlation is visible. Is this still correct (due to the high n number?)

7) Page #15: In Fig 4D…. Is this not 4F?

8) Fig. 4B: Why is in model 2 the pre-mRNA higher expressed then the mRNA?

9) In Figure 5 the authors compare rhythmic nuclear RNAs. Based on panel E these also show rhythms in the cytoplasm. Is this with the same acrophase?

10) Based on 4D many NAE rhythmic transcripts show a low amplitude. Could the authors comment how reliably these can be predicted (especially with n=2 I would estimate some false positive genes).

11) The authors mention a spike in rat RNA. Could they comment why this was not used for normalization. Would this result in comparable results?

Reviewer #2: In the manuscript entitled “Comprehensive analysis of the circadian nuclear and cytoplasmic transcriptome in mouse liver”, Hurni et al. generate and analyze new circadian datasets from cellular fractions obtained from mouse liver and determined the kinetics of gene expression. Briefly, the authors extract RNA from nuclear and cytoplasmic fractions of mouse livers collected at different times of the day (every 4 hours) and use it to generate polyA and total RNAseq libraries which they sequence. By plugging the data into several mathematical models, the authors can determine not only genes displaying rhythmic transcription and steady state levels (as done previously by many including the Naef lab) but also degradation, splicing and nuclear export rates. Among the many findings the authors uncover a role for nuclear export in generating and modulating circadian gene expression.

The present manuscript presents both very interesting datasets as well as findings regarding not only circadian gene expression but also how gene expression in general can be modulated at the different levels. While some of the findings might seem only confirmatory, the present study is solid and constitute a body of work that should be published and shared with the scientific community. So, I strongly support publication in PLoS Genetics with minor revisions.

Minor Concerns

- Most of the results are based on the analysis of RNAseq datasets. In this context it would be good if the authors could speculate how the oscillations of the genes in the M4 group is generated.

- The fact that splicing seems less efficient in longer genes is no surprising. Maybe the authors could interpret this in the context of effects of transcriptional elongation on cotranscriptional splicing.

Reviewer #3: Hurni et al. produced time series RNA-seq data sets from fractionated mouse liver cells and used the obtained RNA measurements to build a mathematical model, which allows to investigate RNA processing steps namely nuclear export and cytoplasmic degradation. Furthermore, they also studied the proportion of co-transcriptionally and post-transcriptionally spliced genes through the intronic signals present within nuclear total pre-mRNA and the nuclear polyadenylated pre-mRNA and reported a strong correlation between gene length (using only protein coding transcripts) and the tendency towards co-transcriptional or post-transcriptional splicing.

Overall, I found the idea driving this work to be interesting. Valuable data sets were produced, which will be appreciated by others investigating circadian regulation underlying RNA processing steps. Potentially interesting results, though still preliminary, are presented. I have a few major and minor concerns which should be addressed before publication.

Major concerns:

1- Overall, I found that the main conclusions are often hidden in a description of the methodology. This is often the case in the results section where the authors frequently describe the methods, but not the interpretation of the results presented.

2- In the discussion a summary of the results is presented, but little is mentioned regarding the contextualization of these results within the landscape of published data. This is particularly evident when referring to the splicing findings. The literature on circadian regulation of splicing seems to have been, to a large extend, neglected (e.g. McGlincy NJ, et al. Genome Biol. 2012. doi: 10.1186/gb-2012-13-6-r54;. Benegiamo, G. et al RNA Process; Springer, 2016; pp. 107–122.; El-Athman R, et al. EBioMedicine. 2018. doi: 10.1016/j.ebiom.2018.06.012; Torres M, et al. Wiley Interdiscip Rev RNA. 2018 doi: 10.1002/wrna.1467).

3- How much of the findings apply only to the liver (which is of course fine and to some degree expected)? Alternative splicing is known to show tissue specificity (Baralle, F et al. Nat. Rev. Mol. Cell Biol. 2017) and also circadian regulation seems to be tissue specific (Litovchenko M, et al. Sci Adv. 2021 Jan 29;7(5):eabc3781. doi: 10.1126/sciadv.abc3781.; Harder L, et al. Bioessays. 2020 May;42(5):e1900158. doi: 10.1002/bies.201900158.). Can the authors use published mammalian circadian data sets to verify some of their conclusion, or to compare the outcomes using different tissues?

4- Related to the previous comment: it is relevant to specify, for the different conclusion, that the current findings apply only to liver tissues. I assume that the RNAseq data is submitted to a public repository, however this is not indicated nor is an accession number provide, which could be used to retrieve the data, after publication of the manuscript.

5- Some references are missing, for e.g. the authors write “…to couple circadian genes and physiological pathways.”, pg 3, a reference would be needed.

6- The authors present a circadian analysis based on the expression of the genes, yet often the individual transcripts oscillate, but when summarizing their expression, the result is a non-oscillating gene (expression), this of course will influence the conclusions and should be taken into account, in particular considering the underlying splicing analysis.

7- It is often not clear which isoform of a certain gene is being considered and several of the genes mentioned have more than one isoform.

8- How would the authors account for alternative splicing using their model, or are they considering simply one transcript per pre-RNA, and if yes, what would be the rationale to ignore alternative splicing events?

9- The authors refer to “splicing time” how is splicing time determined and how was it validated?

10- The authors refer to Correlation coefficient (r) values in the results section, I could not find a description for the type of correlation computed. Overall the information regarding statistics is lacking throughout the manuscript. The authors show (e.g. Fig 4) potentially oscillating genes, but do not provide p-values/adjusted p-values for the significance of such oscillation. Also in Fig 4F in some plots an error for the half-life values is provided and in some not. The information regarding what type of error this is (e.g. SD, SEM) is missing, as well as which N was used.

11- In Fig 6 D, F it is not clear how many nuclei were used for the quantification, statistics are also missing to evaluate whether the apparent differences observed are significant.

12- Which splicing factors are the authors considering for their analysis? For e.g. in Fig 2 the authors refer to “splicing (SR family members)”, but this would be a rather reductionist view of the group of splicing factors, it should be made clear in the manuscript which SFs are being considered.

13- The authors mentioned changes in the circadian profile of splicing factors such as Hnrnph1 or Srsf1, where alterations in spliceosome complex elements such as U2af1 detected?

14- In the results section, the authors mentioned that “The splicing regime is strongly influenced by the gene length, or more precisely by the relation of splicing and elongation time”. However, the authors have only focused on the transcripts with protein coding biotypes during this analysis, thus it should be specified that this conclusion is only valid for such transcripts.

15- What was the rationale to focus on PARBZip family members?

16- Have the authors found a correlation between the number of transcripts (or maybe number of circadian transcripts) from a given gene and co- vs post-transcriptional splicing?

Minor concerns:

• In the Methods section, the catalogue numbers, company names, and locations are missing.

• Figure 1, panel E: It is currently difficult to differentiate between dashed and solid lines (exon and introns).

• There are few typos in the manuscript e.g. “bioype”, “focussing”.

• The magnification details for the microscopy pictures is missing (Figure 6).

• Missing axis labels in Figure 6A, Figure S2.

• In the Microscopy and image analysis, missing “analysis” in the sentence “the image was performed using ImageJ”.

• Missing details on ImageJ and microscope software version.

Overall, the study design and purpose are interesting, but a better description and interpretation of the results is needed to highlight the major outcomes, as well as improving the contextualization and association of ideas in the discussion.

**Have all data underlying the figures and results presented in the manuscript been provided?**

Reviewer #1: **No: **RNA sequencing data are not publically available

Reviewer #2: Yes

Reviewer #3: **No: **RNA seq data should be deposited in a public repository

PLOS authors have the option to publish the peer review history of their article (what does this mean?). If published, this will include your full peer review and any attached files.

Reviewer #1: No

Reviewer #2: No

Reviewer #3: No

---

## [Decision Letter · Decision Letter 1]

3 May 2022

Dear Felix,

Thank you very much for submitting your Research Article entitled 'Comprehensive analysis of the circadian nuclear and cytoplasmic transcriptome in mouse liver' to PLOS Genetics. The revised manuscript was seen by reviewer #3. They (and we) are enthusiastic about the work and are prepared to move forward with acceptance. However, there are some minor concerns from reviewer #3 that we ask you to address in a final round of minor revision that should not require additional external review.

[LINK]

Yours sincerely,

Achim Kramer

Associate Editor

PLOS Genetics

Gregory Barsh

Editor-in-Chief

PLOS Genetics

Reviewer's Responses to Questions

**Comments to the Authors:**

Reviewer #3: Hurni et al. have revised the manuscript following the recommendations of the reviewers. I have carefully read the rebuttal letter and the replies to the comments of the three reviewers, which I found clear and convincing. I would also like to praise the authors for carrying out additional analysis and introducing several clarifications and additional discussion.

This is a very interesting work, which fully deserves publication in PLOS Genetics. I have though a few minor comments, all related to discrepancies between the rebuttal letter and the revised version of the manuscript, which the authors can easily solve.

I fully agree with all the replies from the authors in the rebuttal letter, however, for some of the points, I could not find, in the version of the manuscript with corrections marked, the corresponding additions/clarifications, or not as clear and as detailed as in the letter. Thus, I would ask the authors to carefully compare rebuttal vs manuscript for the points listed below where I use the same numbering as in the rebuttal letter.

2- In the discussion a summary of the results is presented, but little is mentioned regarding the contextualization of these results within the landscape of published data. This is particularly evident when referring to the splicing findings. The literature on circadian regulation of splicing seems to have been, to a large extend, neglected (e.g. McGlincy NJ, et al. Genome Biol. 2012. doi: 10.1186/gb-2012-13-6-r54;. Benegiamo, G. et al RNA Process; Springer, 2016; pp. 107–122.; El-Athman R, et al. EBioMedicine. 2018. doi: 10.1016/j.ebiom.2018.06.012; Torres M, et al. Wiley Interdiscip Rev RNA. 2018 doi: 10.1002/wrna.1467).

Reply: (…) Nevertheless, we agree that a better discussion of the literature on circadian alternative splicing was necessary and have now included the discussion of the listed references.

8- How would the authors account for alternative splicing using their model, or are they considering simply one transcript per pre-RNA, and if yes, what would be the rationale to ignore alternative splicing events?

Reply: (…) The fact that different isoforms may have different rhythmic patterns that would be hidden at the gene-level is now clearly mentioned in the discussion, and we referenced relevant papers (El-Athman et al. 2019).

9- The authors refer to “splicing time” how is splicing time determined and how was it validated?

Reply: (…). These points have been clarified in the discussion section, as also suggested under points 2 and 3 from this reviewer.

**Have all data underlying the figures and results presented in the manuscript been provided?**

Reviewer #3: Yes

PLOS authors have the option to publish the peer review history of their article (what does this mean?). If published, this will include your full peer review and any attached files.

Reviewer #3: No

---

## [Editor Report · Decision Letter 2]

6 Jul 2022

Dear Felix,

We are pleased to inform you that your manuscript entitled "Comprehensive analysis of the circadian nuclear and cytoplasmic transcriptome in mouse liver" has been editorially accepted for publication in PLOS Genetics. Congratulations!

Yours sincerely,

Achim Kramer

Associate Editor

PLOS Genetics

Gregory Barsh

Editor-in-Chief

PLOS Genetics

Comments from the reviewers (if applicable):

**Data Deposition**

http://datadryad.org/submit?journalID=pgenetics&manu=PGENETICS-D-21-01403R2

**Press Queries**

---

## [Editor Report · Acceptance letter]

31 Jul 2022

PGENETICS-D-21-01403R2 

Comprehensive analysis of the circadian nuclear and cytoplasmic transcriptome in mouse liver 

Dear Dr Naef, 

We are pleased to inform you that your manuscript entitled "Comprehensive analysis of the circadian nuclear and cytoplasmic transcriptome in mouse liver" has been formally accepted for publication in PLOS Genetics! Your manuscript is now with our production department and you will be notified of the publication date in due course.

With kind regards,

Olena Szabo

PLOS Genetics

On behalf of:
